# Privacy-preserving Learning via Deep Net Pruning

## Abstract

Neural network pruning has demonstrated its success in significantly improving the computational efficiency of deep models while only introducing a small reduction on final accuracy. In this paper, we explore an extra bonus of neural network pruning in terms of enhancing privacy. Specifically, we show a novel connection between magnitude-based pruning and adding differentially private noise to intermediate layers under the over-parameterized regime. To the best of our knowledge, this is the first work that bridges pruning with the theory of differential privacy. The paper also presents experimental results by running the model inversion attack on two benchmark datasets, which supports the theoretical finding.

## 1 Introduction

Data privacy has become one of the top concerns in the application of deep neural networks, since there has been an increasing demand to train deep models on private data sets. For example, hospitals are now training their automated diagnosis systems on private patients' data Litjens et al. (2016); Lakhani & Sundaram (2017); De Fauw et al. (2018); and advertisement providers are collecting users' online trajectories to optimize their learning-based recommendation algorithm Covington et al. (2016); Ying et al. (2018). These private data, however, are usually subject to the regulations such as California Consumer Privacy Act (CCPA) Legislature (2018), Health Insurance Portability and Accountability Act (HIPAA) Act (1996), and General Data Protection Regulation (GDPR) of European Union.

Differential privacy (DP) Dwork et al. (2006b); Dwork (2009); Dwork & Roth (2014) has emerged, during the past few years, as a strong standard to provide theoretical privacy guarantees for algorithms on aggregate databases. The core idea of achieving differential privacy is to add controlled noise to the output of a deterministic function, such that the output cannot be used to infer much about any single individual in the database. Recent years have seen an increasing number of applications that adapt differential privacy mechanisms to address privacy concerns in deep learning Shokri & Shmatikov (2015); Abadi et al. (2016); Phan et al. (2016); McMahan et al. (2018).

Neural network pruning (or pruning in short), a seemingly orthogonal field to privacy, has also been the subject of a great amount of work in recent years. Pruning aims to reduce the number of model parameters, such that the compressed model can be applied even under the memory constraints of the edge-devices. Various pruning techniques have succeeded in significantly compressing models with little or no loss of accuracy Han et al. (2015; 2016a); Li et al. (2016); Ding et al. (2018); Evci et al. (2019); Tanaka et al. (2020). However, the majority of existing literature only demonstrate the benefits of pruning in terms of energy saving and inference speedup, while in this work, we investigate another interesting bonus of pruning – *preserving data privacy.*

Our investigation is mainly inspired by the observation that neural network pruning makes the inversion from hidden-layers harder, as the percentage of remained weight decreases (see Figure 1). Motivated by this empirical observation, we build under the over-parmeterized regime of deep learning theory, and show an interesting connection between neural network pruning and adding differentially private noise to intermediate layers. We believe this con-

Figure 1: Visualization of inverted CIFAR-10 Krizhevsky (2009) inputs from the third bottleneck of ResNet-18 He et al. (2016) using the inversion algorithm in Section 5.2. We prune the network with different $k$'s, the fraction of remained weights. Inverted images from networks with fewer weights are visually more different from the original image.

nection may have important practical implications since the pruned model only incurs small accuracy loss, and we leave that as future work.

We list our contributions as follow:

- We explore the benefits of pruned neural networks in terms of preserving data privacy. To the best of our knowledge, this is the first step towards drawing a theoretical connection between neural network pruning and differential privacy.
- To build the connection between pruning and adding differentially private noise to intermediate layers, we generalize the famous anti-concentration inequality due to Carbery and Wright Carbery & Wright (2001). This generalization might find more applications in the theoretical analysis of neural network pruning in the future.
- We provide empirical results in support of our theoretical finding. Specifically, we demonstrate on two benchmark datasets that pruned neural networks are more secure in the sense that running the model inversion attack becomes harder.

**Roadmap.** The rest of this paper is organized as follow. Section 2 covers existing literature in different privacy, neural network pruning, and over-parameterized deep learning theory. Section 3 provides theoretical preliminaries and Section 4 presents our main theoretical result. Section 5 shows empirical results on MNIST and CIFAR-10 benchmarks that are in support of our theoretical findings. We conclude this work in Section 6.

## 2 RELATED WORK

**Neural network pruning** Traditional deep neural network models are computationally expensive and memory intensive, which hinders their deployment in applications with limited memory resources or strict latency requirements. Many progress has been made to perform model compression in deep networks, including low-rank factorization Sainath et al. (2013); Lebedev et al. (2015), network pruning LeCun et al. (1990); Srinivas & Babu (2015); Han et al. (2016b); Li et al. (2017), and knowledge distillation Hinton et al. (2015); Chen et al. (2017). Among them, neural network pruning has been widely adopted because it is able to reduce model sizes by up to one order of magnitude without significant accuracy loss. The idea of network pruning dates back to the Optimal Brain Damage in 1990s LeCun et al. (1990). Recently, it has been shown that removing the weights with low magnitude can also achieve a highly compressed model Han et al. (2016b), which is referred to as 'magnitude-based pruning'.

**Differential privacy** The concept of $\epsilon$-differential privacy was originally introduced by Dwork, McSherry, Nissim and Smith Dwork et al. (2006b). Later, it was generalized to a relaxation of $(\epsilon, \delta)$-differential privacy Dwork et al. (2006a); Dwork (2009); Dwork & Roth (2014). Differential privacy has been successfully applied to many problems. For more detailed surveys of the applications of differential privacy, we refer the readers to Dwork (2008; 2011). Applying differential privacy techniques in deep learning is an interesting but non-trivial task. Previous research have customized differential privacy for different learning tasks and settings Shokri & Shmatikov (2015); Abadi et al. (2016); Phan et al. (2016).

Although there are existing studies about applying differential privacy in neural network, but there is little exploration on presenting differential privacy using prune network yet.

To the best of our knowledge, this paper is the first work that shows a connection between differential privacy and pruned neural network.

**Over-parameterized deep learning theory** Recently, there is super long line of work focusing showing the convergence of deep neural network training under over-parameterization regime Li & Liang (2018); Du et al. (2019); Allen-Zhu et al. (2019b;c;a); Arora et al. (2019a;b); Song & Yang (2019); Oymak & Soltanolkotabi (2020); Brand et al. (2020). The theory suggested as long as the neural network is sufficiently wide, i.e., $m \geq \mathrm{poly}(n, d, 1/\delta, L)$ then running (stochastic) gradient descent algorithm is able to find the global minimum, where $n$ is the number of input data points, $d$ is the dimension of data, $\delta$ is the minimum $\ell_2$ distances between all pairs and $L$ is the number of layers in neural network.

However, unlike the above classical deep learning convergence theory, this work explored the over-parameterized theory in a very different perspective, e.g. privacy. Our result is not an optimization result which indicating neural network can learn a set of input data points in certain sense, however our result is suggesting neural network can be private in the differential privacy sense.

## 3 Backgrounds

**Notations.** For a positive integer $n$, we use $[n]$ to denote set $\{1, 2, \cdots, n\}$. For vector $x \in \mathbb{R}^n$, we use $\|x\|_1$ to denote $\sum_{i=1}^n |x_i|$, $\|x\|_2$ to denote $(\sum_{i=1}^n x_i^2)^{1/2}$, $\|x\|_\infty$ to denote $\max_{i \in [n]} |x_i|$. We use $\mathcal{N}(\mu, \sigma^2)$ to denote random Gaussian distribution. For a matrix $A$, we use $\|A\|$ to denote its spectral norm.

This section presents some backgrounds before theoretically establishing the equivalence between magnitude-based pruning and adding differentially private noise in Section 4. Section 3.1 revisits the notion of $(\epsilon_{\mathrm{dp}}, \delta_{\mathrm{dp}})$-differential privacy. Section 3.2 describes the magnitude pruning algorithm.

### 3.1 Differential privacy

The classical definition of differential privacy is shown as follow:

**Definition 3.1** (($(\epsilon_{\mathrm{dp}}, \delta_{\mathrm{dp}})$-differential privacy Dwork et al. (2006a)). *For a randomized function $h(x)$, we say $h(x)$ is $(\epsilon_{\mathrm{dp}}, \delta_{\mathrm{dp}})$-differential privacy if for all $S \subseteq \mathrm{Range}(h)$ and for all $x, y$ with $\|x - y\|_1 \leq 1$ we have*

$$\Pr_h[h(x) \in S] \leq \exp(\epsilon_{\mathrm{dp}}) \cdot \Pr_h[h(y) \in S] + \delta_{\mathrm{dp}}.$$

Definition 3.1 says that, if there are two otherwise identical records $x$ and $y$, one with privacy-sensitive information in it, and one without it, and we normalize them such that $\|x - y\|_1 \leq 1$. Differential Privacy ensures that the probability that a statistical query will produce a given result is nearly the same whether it's conducted on the first or second record. Parameters $(\epsilon_{\mathrm{dp}}, \delta_{\mathrm{dp}})$ are called the privacy budget, and smaller $\epsilon_{\mathrm{dp}}$ and $\delta_{\mathrm{dp}}$ provide a better differential privacy protection. One can think of a setting where both parameters are 0, then the chance of telling whether a query result is from $x$ or from $y$ is no better than a random guessing.

A standard strategy to achieve differential privacy is by adding noise to the the original data $x$ or the function output $h(x)$. In order to analyze it, we need the following definition:

**Definition 3.2** (Global Sensitivity Dwork et al. (2006b)). *Let $f : \mathbb{R}^n \to \mathbb{R}^d$, define $\mathrm{GS}_p(f)$, the $\ell_p$ global sensitivity of $f$, for all $x, y$ with $\|x - y\|_1 \leq 1$ as*

$$\mathrm{GS}_p(f) = \sup_{x, y \in \mathbb{R}^n} \|f(x) - f(y)\|_p.$$

The global sensitivity of a function measures how 'sensitive' the function is to slight changes in input. The noise needed for differential privacy guarantee is then calibrated using some

well-known mechanisms, e.g., Laplace or Gaussian Dwork & Roth (2014), and the amount of noise (the standard deviation of the noise distribution) is proportional to the sensitivity, but inversely proportional to the privacy budget $\epsilon_{\mathrm{dp}}$. That is to say, for a given function with fixed global sensitivity, a larger amount of noise is required to guarantee a better differential privacy (one with a smaller budget $\epsilon_{\mathrm{dp}}$).

## 3.2 MAGNITUDE-BASED PRUNING

Magnitude-based pruning Han et al. (2016a) compresses a neural network by removing connections with smallest-magnitude weights (usually determined by using a threshold, say $\alpha$). The pruning procedure starts with a trained (dense) network $f(W_1, \cdots, W_L)$, where $L$ is its depth, and $W_l, l \in [L]$ is the weight of its $i$-th layer. For each layer $W_l$, it sets the weights with magnitudes smaller than $a$ to zero:

$$(W_l)_{i,j} \leftarrow (W_l)_{i,j} \cdot \mathbf{1}_{|(W_l)_{i,j}|>a}, \forall i,j$$

We then run model update after the pruning step.

## 4 MAIN RESULT

We start by formulating the connection between pruning and adding differentially private noise. We define the following notions to describe the closeness between a randomized function and a given function, which can be either randomized or deterministic.

**Definition 4.1** (($\epsilon_{\mathrm{ap}}, \delta_{\mathrm{ap}}$)-close). *For a pair of functions $g : \mathbb{R}^d \to \mathbb{R}^m$ and $h : \mathbb{R}^d \to \mathbb{R}^m$, and a fixed input $x$, we say $g(x)$ is $(\epsilon, \delta)$-close to $h(x)$ if and only if,*

$$\Pr_{g,h} \left[ \frac{1}{\sqrt{m}} \|g(x) - h(x)\|_2 \leq \epsilon \right] \geq 1 - \delta.$$

($\epsilon_{\mathrm{ap}}, \delta_{\mathrm{ap}}$)-closeness basically requires that, $\ell_2$ distance between two functions' output with a given input is small enough ($\|f - g\|_2 := (\int_x |f(x) - g(x)|^2 \mathrm{d}x)^{1/2}$). When applying the definition to deep neural network, we view $m$ as the width of deep neural network (e.g. the number of neurons).

Let $\phi(t)$ denote the activation function. In this work, we focus on ReLU case where $\phi(t) = \max\{t, 0\}$. Without loss of generality, our techniques can be generalized to other activation functions. We present our main theoretical result in Theorem 4.2. The main message of our theorem is that, as long as the neural network is sufficiently wide, pruning neural network has a similar effect to adding differentially private noise.

**Theorem 4.2** (Main result, informal of Theorem E.1). *Let $\sigma_A = O(\epsilon_{\mathrm{dp}}\delta_{\mathrm{dp}}/(m^2))$. For a fully connected neural network (each layer can be viewed as $f(x) = \phi(Ax + b)$), where $\|x\|_2 = 1$ and $x \in \mathbb{R}^d_{\geq 0}$. Applying magnitude-based pruning on the weight $A \in \mathbb{R}^{m \times d}$ (where each $A_{i,j} \sim \mathcal{N}(0, \sigma_A^2)$) gives us $\widetilde{A} \in \mathbb{R}^{m \times d}$.
If $m = \Omega(\mathrm{poly}(1/\epsilon_{\mathrm{ap}}, \log(1/\delta_{\mathrm{ap}}), \log(1/\delta_{\mathrm{dp}})))$, then there exists a function $h(x)$ satisfying two properties :*

1. *$h(x)$ is $(\epsilon_{\mathrm{dp}}, \delta_{\mathrm{dp}})$-differential privacy on input $x$;*

2. *$h(x)$ is $(\epsilon_{\mathrm{ap}}, \delta_{\mathrm{ap}})$-close to $g(x) = \phi(\widetilde{A}x + b)$.*

In the above theorem, we denote $d$ as the input data dimension. $\phi$ is the activation function, e.g., $\phi(z) = \max\{z, 0\}$. In this work, we focus on one hidden layer neural network.[1]

---

[1]We would like to emphasize that one hidden layer is *not* just a toy example, but a natural and standard situation to study theory, see Zhong et al. (2017b;a); Li & Yuan (2017); Li & Liang (2018); Du et al. (2019); Song & Yang (2019); Brand et al. (2020); Bubeck et al. (2020) for example. Usually, if a proof holds for one-hidden layer, generalizing it to multiple layers is straightforward Allen-Zhu et al. (2019b;c)

Regarding the two properties of $h(x)$, property 1 requires $h(x)$ to provide $(\epsilon_{\mathrm{dp}}, \delta_{\mathrm{dp}})$-differential privacy, and property 2 requires that $h(x)$ is similar to magnitude-based pruning with the predefined $(\epsilon_{\mathrm{ap}}, \delta_{\mathrm{ap}})$-close notation.

**Proof Sketch**   Let $\widetilde{A} \in \mathbb{R}^{m \times d}$ denote the weight matrix after magnitude-based pruning, and $\overline{A} = \widetilde{A} - A \in \mathbb{R}^{m \times d}$. We define a noise vector $e \in \mathbb{R}^m$ as the follow:

$$e = \mathrm{Lap}(1, \sigma)^m \circ (\overline{A}x).$$

The main proof can be split into two parts in correspondence to property 1 and 2 respectively: Claim 4.3 and Claim 4.4.

**Claim 4.3.** *Let $h(x) = f(x) + e \in \mathbb{R}^m$, we can show that $h(x)$ is*

$$(\epsilon_{\mathrm{dp}}, \delta_{\mathrm{dp}}) - differential \ \ privacy.$$

**Claim 4.4.** *For sufficiently large $m$, we have*

$$\Pr\left[\frac{1}{\sqrt{m}}\|e - \overline{A}x\|_2 \geq \epsilon_{\mathrm{ap}}\right] \leq \delta_{\mathrm{ap}}.$$

**Proof sketch of Claim 4.3**   To prove Claim 4.3, we anchor from the definition of differential privacy. Recall Def. 3.1, $(\epsilon_{\mathrm{dp}}, \delta_{\mathrm{dp}})$-differential privacy requires that for any inputs $x$ and $y$ with $\|x - y\| \leq 1$,

$$\Pr_h[h(x) \in S] \leq \exp(\epsilon_{\mathrm{dp}}) \cdot \Pr[h(y) \in S] + \delta_{\mathrm{dp}}.$$

To be more specific, we use the fact that the noise $e$ is sampled from the Laplace distribution, and try to bound the ratio

$$\frac{p_h(h(x) = t \in S)}{p_h(h(y) = t \in S)}$$

where $p(\cdot)$ denotes the probability density function. To bound the above ratio: first we need to derive and upper-bound the global sensitivity (see Appendix D) of a single-layer neural network.

Recall the well-known anti-concentration result by Carbery & Wright (2001).

**Lemma 4.5** (Carbery and Wright Carbery & Wright (2001)). *Let $p : \mathbb{R}^d \to \mathbb{R}$ denote a degree-$k$ polynomial with $d$ variables. There is a universal constant $C > 0$ such that*

$$\Pr_{x \sim \mathcal{N}(0, I_d)}\left[|p(x)| \leq \delta\sqrt{\mathrm{Var}[p(x)]}\right] \leq C \cdot \delta^{1/k}.$$

Another contribution in this work is that we extend the anti-concentration result Carbery & Wright (2001) to a more general setting, which has not been explored in literature. We state our generalization as follows[2]

**Lemma 4.6** (An variation of Carbery & Wright (2001), Anti-concentration of sum of truncated Gaussians). *Let $x_1, \cdots, x_n$ be $n$ i.i.d. zero-mean Gaussian random variables $\mathcal{N}(0, 1)$. Let $p : \mathbb{R}^n \to \mathbb{R}$ denote a degree-1 polynomial defined as*

$$p(x_1, \cdots, x_n) = \sum_{i=1}^{n} \alpha_i x_i.$$

*Let $f$ denote a truncation function where $f(x) = x$ if $|x| \leq a$, and $f(x) = 0$ if $|x| > a$. Then we have*

$$\Pr_{x \sim \mathcal{N}(0, I_d)}\left[|p(f(x))| \leq \min\{a, 0.1\} \cdot \delta \cdot \|\alpha\|_2\right] \geq C \cdot \delta.$$

Once the densities are bounded, integrating $p(\cdot)$ yields the requirement of differential privacy, thus complete the proof of part 1.

---

[2]for more details of the proof, we refer the readers to Appendix C

**Proof sketch of Claim 4.4**  To prove Claim 4.4, we firstly define $z_i = e_i - (\overline{A}x)_i$. Then we apply the concentration theorem (see Appendix B) to show that for any $\|x\|_2 = 1$ and $x \in \mathbb{R}^d_+$,

$$\Pr\left[\frac{1}{m}|\sum_{i=1}^{m}(z_i - \mathbb{E}[z_i])| \geq \epsilon_{\mathrm{ap}}^2\right] \leq \delta_{\mathrm{ap}},$$

which completes the proof of Claim 4.4.

In certain, the proof is mainly a sophisticated combination of the following concentration inequalities

- Lemm B.1 shows the concentration of the $\ell_2$ norm of a random truncated Gaussian vector
- Lemma B.2 shows the concentration of matrix vector multiplication.
- Lemm B.3 bounds the inner product between a random Guassian vector and a fixed vector
- Lemma B.4 bounds inner product between two random Guassian vectors
- Lemma B.5 shows the concentration of folded Gaussian random vectors.

## 5  EXPERIMENTS

This section presents experimental results in support of our theoretical finding in Section 4 by answering this question: **does the pruned model preserve better privacy than the dense one?** We describe experimental setups in Section 5.1 and the evaluation for privacy in Section 5.2. We summarize results in Section 5.3.

### 5.1  EXPERIMENTAL SETUP

**Datasets and model architectures.**  We have conducted image classification experiments on MNIST LeCun et al. (2010) and CIFAR-10 Krizhevsky (2009) benchmarks.

For network architectures, we have used LeNet-5 LeCun et al. (1998) for MNIST, and ResNet-18 He et al. (2016) for CIFAR-10, with PyTorch Paszke et al. (2019) as the experiment platform. We have used SGD Qian (1999) with learning rate 0.05 and momentum 0.9 for both models, and train LeNet-5 for 20 epochs and ResNet-18 for 150 epochs. All models are trained on 8 NVIDIA GeForce RTX 2080 Ti GPUs with batch size 256.

**Pruning algorithm.**  We have employed the iterative pruning technique, which repeatedly prunes and retrains the network over $n$ rounds: in each round, we prune $p$ fraction of the weights that survive the previous round and retrain for $t$ epochs. We use $k$ to denote the fraction of weights remained in the final sparse model, thus we have $k = (1-p)^n$. In our experiments, we set $p = 20\%$ and $t = 5$, and vary $n \in [15]$ to get pruned networks with different $k$'s.

### 5.2  TEST OF PRIVACY LEAKAGE AS A MODEL INVERSION ATTACK

We have used the attack-based evaluation to investigate whether pruning could preserve more privacy (i.e., suffer less "privacy leakage" under the attack). We have adopted the model inversion attack Mahendran & Vedaldi (2015) to show the privacy leakage of a given $l$-layer neural network $f(W_1, \cdots, W_l)$, where $W_i, i \in [l]$ is the weight of its $i$-th layer.

Let us conceptualize the mapping from the $f$'s input to the output of its $i$-th intermediate as a representation function $\Phi : \mathbb{R}^d \to \mathbb{R}^m$. The model inversion attack captures the potential privacy leakage of $\Phi$ when applied on some input $x$:

Given the representation $\Phi(x)$, and the weights in the public mapping $\Phi$, the attacker's goal is to find the preimage of $\Phi(x)$, namely

$$x^* = \arg\min_{z \in \mathbb{R}^d} \mathcal{L}(\Phi(x), \Phi(z)) + \lambda \mathcal{R}(z)$$

| $k(\%)$ | 100.0 | 80.0 | 64.0 | 51.2 | 41.0 | 32.8 | 26.2 | 21.0 |
|---|---|---|---|---|---|---|---|---|
| MNIST | 99.0 | 98.9 | 99.0 | 99.0 | 99.0 | 99.0 | 99.0 | 98.9 |
| CIFAR-10 | 94.1 | 93.8 | 93.6 | 93.4 | 93.0 | 92.5 | 91.9 | 90.7 |
| $k(\%)$ | 16.8 | 13.4 | 10.7 | 8.6 | 6.9 | 5.5 | 4.5 | 3.5 |
| MNIST | 98.8 | 98.8 | 98.7 | 98.4 | 98.1 | 98.0 | 97.5 | 97.5 |
| CIFAR-10 | 89.8 | 88.2 | 86.3 | 83.3 | 75.9 | 72.4 | 61.9 | 10.0 |

Table 1: Test accuracy (%) achieved on MNIST and CIFAR-10 when pruning a dense network. $k$ is the fraction of weights remained.

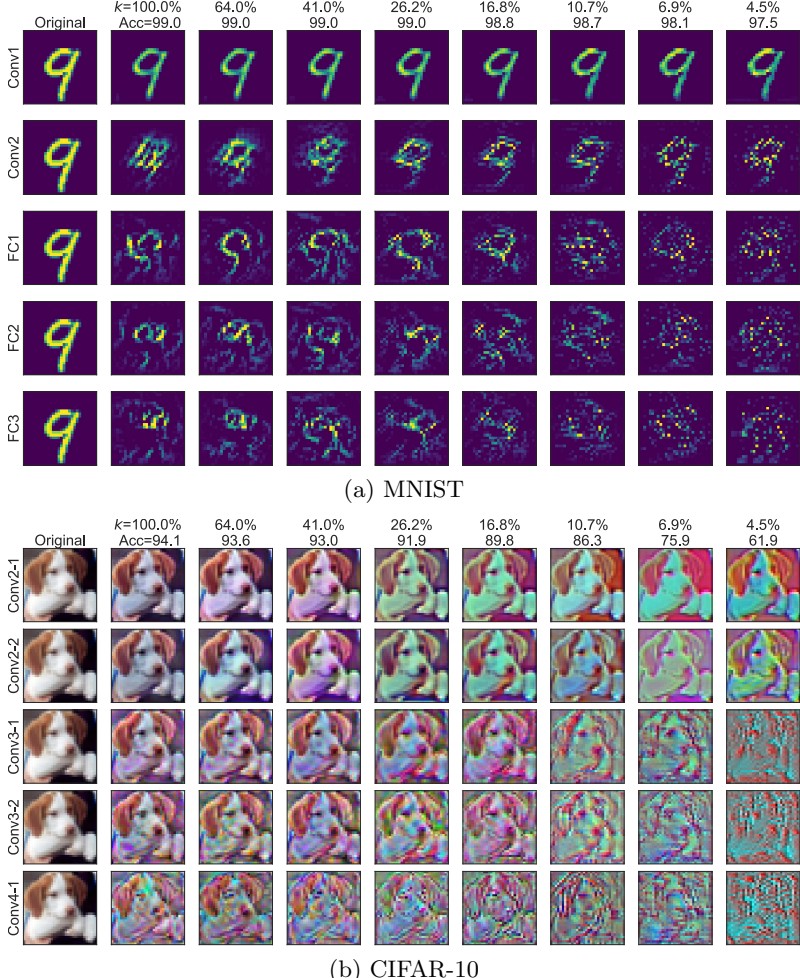

(a) MNIST

(b) CIFAR-10

Figure 2: Recovered MNIST digits (a) and CIFAR-10 samples (b) by running the model inversion attack in Section 5.2 on LeNet-5 and ResNet-18 models trained with pruning. We test different layers (different rows) and fractions of remained weights ($k$, different columns). The naming of layers is explained in Section 5.3.

where the loss function $\mathcal{L}$ is defined as $\mathcal{L}(a, a') = \|a - a'\|_2^2$. $\lambda > 0$ is the regularization parameter, and the regularization function $\mathcal{R}$ in our case is the total variation of a 2D signal: $\mathcal{R}(a) = \sum_{i,j}((a_{i+1,j} - a_{i,j})^2 + (a_{i,j+1} - a_{i,j})^2)^{1/2}$.

## 5.3 RESULTS

We have evaluated the model inversion attack on different representation function $\Phi$'s by

1. Pruning the neural network with different $k$'s, the fraction of remained weights, where $k = (1 - 20\%)^n, n \in [15]$

2. Selecting different intermediate layers to invert. Specifically, for LeNet-5, we run the attack on all 5 layers, namely {'Conv1', 'Conv2', 'FC1', 'FC2', 'FC3'}. For ResNet-18, we pick 5 layers {'Conv2-1', 'Conv2-2', 'Conv3-1', 'Conv3-2', 'Conv4-1'}, where 'Conv$i$-$j$' stands for the $j$-th convolutional group of the $i$-th convolutional block.

Now we report that our experimental results strongly suggest that magnitude-based pruning preserves better privacy (suffer less leakage) than its dense counterpart.

**Accuracy of network pruning.** Table 1 shows the accuracy results of MNIST and CIFAR-10 tasks for the network pruning approach whose networks have different $k$ percentages of remaining weights. The task on MNIST achieves the same level of accuracy as the network model without pruning when $k$, the fraction of remained weights $\geq 10\%$. Its accuracy gradually decreases as more weights get removed. The test with CIFAR-10 maintains the same accuracy as or better than the model without pruning when $k \geq 40\%$, and then gradually decreases as $k$ decreses.

Note that the model accuracy achieved with the pruning algorithm in our experiments may be lower than that of the SOTA model with the same sparsity. However, for answering the question if network pruning preserves privacy, our experiments can be viewed as conservative results.

**Sparser networks suffer less privacy leakage.** Figure 2 visualized the inverted samples under different choices of layers and fractions of remained weights. Each column of Figure 2 suggests the increasing difficulty of inverting deeper layers.

A more important observation is that, for all layers, we consistently observe that the privacy leakage gradually decreases as $k$ increases: for deep layers (e.g. 'FC2' for LeNet-5 and 'Conv4-1' for ResNet-18), though the inverted image from the dense model (i.e. $k = 100\%$) may look quite identical to the original image without pruning, it is no longer true when $k \leq 90\%$). This agrees with Theorem 4.2, which suggests that under the over-parameterized regime, pruning yields a similar effect to adding differentially private noise and thus preserves better privacy.

## 6 CONCLUSIONS

This paper has presented a theoretical result to show that, if a fully-connected layer of a neural network is wide enough, there is a connection between magnitude-based neural network pruning and adding differentially private noise to the model's intermediate outputs. Empirical results on two benchmark datasets support our theoretical findings.

These results have strong practical implications for two reasons. First, since neural network pruning has the property that the fraction of removed weights can be quite high (e.g. $> 90\%$) without reducing inference accuracy, it strongly suggests that network pruning can be an effective method to achieve differential privacy without any or much reduction of accuracy. Second, although the result is for a single layer of a neural network, it is quite natural in a distributed or federated learning system to use a particular layer to communicate among multiple sites.

Several questions remain open. First, Theorem 4.2 is only for a single-layer fully connected network, and it would be interesting if one can extend it to multi-layer settings and also convolutional neural networks. Second, our theoretical finding is based on the worst case analysis, which means in most cases, $m$ can be much smaller. How to efficiently determine $m$ for different settings requires more investigation. Finally, in order to use network pruning as a mechanism to preserve privacy in a practical distributed or federated learning system, one needs to consider many design details including which layers to prune, whether or not to prune layers with the same sparsity, where the work of pruning should be performed, and how to coordinate among multiple sites.

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
