# OpenReview forum: "Privacy-preserving Learning via Deep Net Pruning"
_ICLR.cc/2021/Conference — Reject_

### Official Review · AnonReviewer2 · 2020-10-24
**Review of Privacy-preserving Learning via Deep Net Pruning**

**Rating:** 4
**Confidence:** 4

**Review:**

Overview: This paper aims to establish a theoretical connection between differential privacy and magnitude based pruning. The authors show theoretically that outputs of pruned single layer neural networks have some similarity to outputs of the same network with differential privacy added. The paper then empirically demonstrates that model inversion attacks are harder against magnitude pruned neural networks.

Reason for score:
I tend to vote reject. Overall, I think this paper contains several correct and interesting pieces that are put together the wrong way. The empirical observation that pruned networks are harder to invert is very supported by experiments in this paper; the main theorem 4.2 is (I assume) correctly derived with some new theoretical tools that may be of independent interest. Yet, I am unconvinced about the relation between pruning and differential privacy claimed by this paper due to the concerns that listed below.

Pros:
Paper is well written and mostly concise in presentation. The experiments are convincing in supporting the claim that pruned networks are harder to invert. I have glanced through the supplementary material and the derivations seem correct.

Cons:
1. Statement about the definition of differential privacy is inconsistent: In definition 3.1, the authors define differential privacy with respect to pairs of real vectors x and y with l1 norm less than 1. This definition is incorrect as differential privacy is defined on pairs of databases (set/multiset of vectors or strings), and the neighbouring relationship between pairs of databases should be those which differ by at most one entry. In contrast, the authors appear to use a different definition in appendix D, where x and y now differ by “one entry”, which I presume to mean in one dimension of the vector. This would be analogous to protecting the value of one pixel in a database of a single image, which is a rather useless notion of privacy.

2. Assumption on weights: the main contribution of this paper, which is theorem 4.2, assumes that the network weights are iid drawn from a normal distribution. This assumption seems far fetched since trained neural network weights tend to be strongly correlated rather than independent. I am also not aware of any empirical work who states that magnitudes of trained neural network weights are well approximated by a gaussian. Hence, I am not convinced that the result drawn in theorem 4.2 has real applicability to understanding the relation between DP and pruning (if any such relation exists at all).

3. Conclusion drawn from theorem 4.2: while h(x) satisfies authors’ definition of DP (I discussed by why this definition is problematic in comment 1) and is close to g(x), this should be interpreted as neither adding noise or applying pruning significantly deviates from f(x) in probability. The authors instead use 4.2 to draw similarities between adding noise and applying pruning, which I think is not well supported by their findings.

4. Relationship between Pruning and DP: In DP, the mechanism is required to be stochastic in order to maintain privacy given repeated usage. In contrast, unless the original network is re-trained for each input during inference, the output is deterministic. The idea that we can somehow achieve DP (with respect to inputs during inference) by passing the input through a pruned network is thus unrealistic.

Questions:
If we replaced \bar{A}x with another matrix B that has upper and lower bounded spectral norm, wouldn’t it be possible to achieve the similar results to claim 4.3 and 4.4? Namely, let e=Lap^m * Bx, then f(x)+e is “private” and ||e - Bx|| is epsilon small?

[Post rebuttal] The rebuttal has not address any of my main concerns, so my rating stays.

---

> ### Author Response · Authors · 2020-11-25
> **Response to R2**
>
> We would like to thank the reviewers for their constructive comments that led to the improvement of our manuscript. The question about replacing \bar{A}x with another matrix B is an interesting question but it is beyond the scope of our paper. We believe our work is still valuable as the first step to connect pruning and DP.

---

### Official Review · AnonReviewer4 · 2020-10-26
**Novel connections between differential privacy and pruning neural networks. Recommendation: Weak reject.**

**Rating:** 5
**Confidence:** 5

**Review:**

This paper seeks to protect privacy of the dataset involved in learning process. Specifically, it tries to establish some connection between privacy obtained from pruning neural networks and differential privacy. To do this, it generalises the famous anti-concentration inequality by Carbery and Wright. It also provides empirical results to support its empirical findings.

Strengths: Strong mathematical foundation behind the claims. They manage to establish some connection with DP.

Weaknesses: Has the following drawbacks.
-They seem to overestimate the power of their results.
-The connection that they establish doesn't provide a path to actually achieving DP. The connection is an L2 norm connection with the output of a possible DP algorithm. That doesn't say much about privacy.
-The only attack they consider is model inversion attack. What other kinds of attacks is it robust to?
-On page 8, they claim that for CIFAR-10, the accuracy gradually declines after k=40%. That is just not true. It drops by 30%. So, getting actual privacy is very difficult if accuracy is a desired consequence, too.
-Their methods only work on 1-layered NN. Do they generalise? Or any intuition to why they would generalise?

I feel that the negatives outweigh the positives here.

---

> ### Author Response · Authors · 2020-11-25
> **Response to R4**
>
> Thanks for your valuable suggestions. We agree with R4  that the evaluation of more comprehensive attacks will be interesting. Inversion attacks have been used to evaluate privacy under a differential privacy framework [1]. We use inversion attacks for privacy leakage evaluation as a straightforward understanding of the pruning ratio would affect the data reconstruction results, to demonstrate pruning is a fashion to preserve privacy.  We are delighted to consider generalizing the theory to more general cases.
>
> [1] Park, Cheolhee, Dowon Hong, and Changho Seo. "An Attack-Based Evaluation Method for Differentially Private Learning Against Model Inversion Attack." IEEE Access 7 (2019): 124988-124999.

---

### Official Review · AnonReviewer1 · 2020-10-27
**Interesting problem and approach, superficial experimentation and validation**

**Rating:** 4
**Confidence:** 4

**Review:**

My summary of the papers main goal:

The paper aims at drawing a connection between neural network pruning, and privacy of the trained model. More specifically, the authors try to draw similarities between differentially private training of DNNs and pruning. They define a notion of $(\epsilon, \delta)$ closeness, to measure how close a function is to a randomized function, and then they show that magnitude-based pruning is close to randomized addition of noise, for large number of parameters in the network. To experimentally evaluate this, they mount a model inversion attack on two pruned set of DNNs.

Pros:
+ The problem they address is interesting and novel. It could help get more out of DNNs and hit two birds (privacy and compression) with one stone. It could also offer some neat insights.

+ I am not a theory person myself, but the proof sketch for the closeness of DP and pruning seems sound. However, I wonder how this approximation effects the individuals' privacy in practice.

Cons:

- I am very skeptical about the experimental setup. Here are my main concerns:

a) Since the claim in Section 4 is that pruning is close to differential privacy, the experiments in section 5 should also try and depict that. So ideally, what we should see in the experiments is a one-to-one comparison of a DNN trained with DP-SGD, with a DNN pruned using magnitude-based pruning. I think these two sets of models should be compared in terms of accuracy, and their resistance to different set of attacks, such as membership inference and attribute inference/model inversion. However, we do not see a DP-trained model in the results, so we cannot have a conclusive comparison. The authors could use libraries such as Opacus or TF-privacy to train DP models so I think seeing results on that would be interesting.

b) I think a membership inference attack would be a better measure of how close a method is to DP, than model inversion. Based on my own experience w/ differentially private DNNs and both model-inversion and membership inference attacks, I have seen that the notion of privacy that DP protects is close to membership, as reflected in its formulation, than to model inversion. Many average-case privacy mitigations offer protection against model inversion for some samples, but would not be resistant to membership inference attacks.  If DP is to be contended here, mounting membership inference on both a DP and a pruned model would be a better test of privacy, in my opinion.

c) The model inversion attack is also lacking. Visualizing two sets of examples does not really prove anything, especially not DP. DP offers a stringent "worst-case" guarantee. This cannot be asserted or evaluated with two image reconstructions. A better evaluation would be to somehow offer a success rate for the attack, and also provide the lowest and highest score per sample. Maybe there is one example that is actually completely reconstructable. Even if out of 10,000 samples, one is reconstructed, the guarantee is voided.

- Apart from the points mentioned above, I am also having some difficulty drawing parallels between pruning and DP-SGD.  From the explanation in the second paragraph of Section 5, it seems as though pruning happens once every $t$ epochs, and then normal re-training takes place. If that is the case, does the closeness still hold? DP-SGD requires that no data is fed in the model without applying noise to the gradients (unless it is public data). If in the case of pruning there are epochs where the pruning isn't happen, then are we really achieving guarantees?

---

> ### Author Response · Authors · 2020-11-25
> **Response to R1**
>
> We thank R1 for the careful review. Here we clarify the raised concerns regarding our experiments.
>
> -Compare with DP.
>
> The main message of our work is pruning can connect to privacy and we theoretically show the connection between DP and pruning the weights based on magnitude under random gaussian weights assumption. Experimentally, it is not trivial to fairly compare DP with pruning on accuracy as the privacy budgets are not easy to match in practice. Also, as DP cares about the worst-case scenario, comparing a single sample may not be convincing due to the possible randomness in data parcellation and inversion attack algorithm.
>
> -Include membership inference attack.
>
> Thanks for your suggestion. We agree that the evaluation of membership inference attacks will be valuable. We use inversion attacks for privacy leakage evaluation as a straightforward understanding of the pruning ratio would affect the data reconstruction results, to demonstrate pruning is a fashion to preserve privacy. Besides, inversion attacks have been used to evaluate privacy under a differential privacy framework [1].
>
> We would thank R1 again for the valuable feedback that led to the improvement of our manuscript.
>
> [1] Park, Cheolhee, Dowon Hong, and Changho Seo. "An Attack-Based Evaluation Method for Differentially Private Learning Against Model Inversion Attack." IEEE Access 7 (2019): 124988-124999.

---

### Official Review · AnonReviewer3 · 2020-10-31
**No meaningful privacy implications, poor writing**

**Rating:** 2
**Confidence:** 5

**Review:**

### Summary

The paper claims to exhibit a connection between differential privacy and neural network pruning. The main result is that, for a single layer network with Gaussian weights that depend on the privacy parameters, there exists a differentially private function which approximates the pruned network in $L_2$ distance. This is proved by bounding the sensitivity of the network and adding Laplace noise so that the resulting function satisfies differential privacy and approximates the function defined by the pruned network. According to the authors, these results mean that “ network pruning can be an effective method to achieve differential privacy”

### Evaluation

There are multiple serious conceptual issues with this work. Let me list them in order of severity:

* The fact that a differentially private function h approximates another function g has no implications about the extent to which g preserves privacy. Differential (or any) privacy is not a notion that’s in any sense continuous with $L_2$ distance. For example, suppose we have numerical data consisting of n reals $x_1, \ldots, x_n$ in $[0,1]$, and $g = (x_1 + \ldots + x_n)/n$ is their average, and let $h(x)= g(x) + e$ where e is $\mathrm{Lap}(0,1/\varepsilon)$ noise. Then g and h are very close and, in the authors’ terminology, the average “has a similar effect to adding differentially private noise”. Nevertheless, averages are far from being privacy preserving: for example the exact value can be encoded in the lower order bits of the $x_i$ in a way that is preserved by the average, so that the entire data set $x_1, \ldots, x_n$ can be read off from $g(x)$. Note that this example is *exactly* analogous to what the authors prove, just simpler.

More concisely, the authors have gotten things backwards: finding a differentially private approximation h to a function g does not mean that g is in any way privacy preserving.

* The privacy model used here is peculiar, and it is not clear to me when it makes sense to use it. The authors show how to release a *single prediction* of the network on a *single data point* with differential privacy, where differential privacy is defined with respect to two data points being neighboring if they differ in a single coordinate (in the appendix) or have $\ell_1$ norm at most 1 (in the body of the paper). So what is the data of an individual here? For example, if $x$ is an image, are we preserving “pixel privacy”? And why is it interesting to release a single prediction rather than an entire network, as is more common?

* The theorem is proved for random Gaussian weights, and the variance of the weights depend linearly on the privacy parameters. In particular, the variance of the weights depends linearly on $\delta_{dp}$, which is a parameter that’s often assumed to be cryptographically small. In that case, it looks like the network’s output would contain essentially no signal about its input.

* The paper is badly written. There are problems with phrasing (e.g. algorithms are called “differential privacy” rather than “differentially private”), with undefined notation (what is $\circ$ in the proof sketch of the main theorem?), and with inconsistent definitions (neighborhood definitions are different in the body and the appendix), and unexplained assumptions and model choices (the Gaussian weights, the semantics of the privacy model).

---

> ### Author Response · Authors · 2020-11-25
> **Response to R2**
>
> We appreciate R2's feedback and time. We clarify the question raised regarding our theoretical analysis as follows. \circ denotes Hadamard product, for a vector z= x \circ y , where z_i = x_i y_i. We agree with the reviewer. We will change "differential privacy" to "differentially private."  We agree that several definitions, assumptions need to be explained in a better way. We will improve our manuscript in the next version.

---

### Decision · Program_Chairs · 2021-01-07
**Final Decision**

**Decision:**

Reject

**Comment:**

The paper claims to draw a connection between pruning and differential privacy. There seem to be conceptual issues with the paper, highlighted by all reviewers (see particularly Reviewer 3's review), which the authors had no response to. For example, a function approximating another does not imply any transfer of differential privacy. This is a fundamental issue that the authors would need to address.

Some papers (suggested by the area chair and reviewers) related to differential privacy and pruning that the authors may wish to be aware of include https://arxiv.org/abs/1503.02031 and https://arxiv.org/abs/2008.13578